# In Silico Study of the Mechanisms Underlying the Action of the Snake Natriuretic-Like Peptide Lebetin 2 during Cardiac Ischemia

**DOI:** 10.3390/toxins14110787

**Published:** 2022-11-11

**Authors:** Hinda Allaoui, Nedra Rached, Naziha Marrakchi, Ameur Cherif, Amor Mosbah, Erij Messadi

**Affiliations:** 1Plateforme de Physiologie et Physiopathologie Cardiovasculaires (P2C), Laboratoire des Biomolécules, Venins et Applications Théranostiques (LR20IPT01), Institut Pasteur de Tunis, Université Tunis El Manar, Tunis 1068, Tunisia; 2Laboratory of Biotechnology and Bio-Geo Resources Valorization (LR11ES31), Higher Institute of Biotechnology of Sidi Thabet (ISBST), University of Manouba, Tunis 2010, Tunisia

**Keywords:** natriuretic peptides, receptors, cardioprotection, molecular docking, molecular dynamics simulation, virtual screening

## Abstract

Lebetin 2 (L2), a natriuretic-like peptide (NP), exerts potent cardioprotection in myocardial infarction (MI), with stronger effects than B-type natriuretic peptide (BNP). To determine the molecular mechanisms underlying its cardioprotection effect, we used molecular modeling, molecular docking and molecular dynamics (MD) simulation to describe the binding mode, key interaction residues as well as mechanistic insights into L2 interaction with NP receptors (NPRs). L2 binding affinity was determined for human, rat, mouse and chicken NPRs, and the stability of receptor–ligand complexes ascertained during 100 ns-long MD simulations. We found that L2 exhibited higher affinity for all human NPRs compared to BNP, with a rank preference for NPR-A > NPR-C > NPR-B. Moreover, L2 affinity for human NPR-A and NPR-C was higher in other species. Both docking and MD studies revealed that the NPR-C–L2 interaction was stronger in all species compared to BNP. Due to its higher affinity to human receptors, L2 could be used as a therapeutic approach in MI patients. Moreover, the stronger interaction of L2 with NPR-C could highlight a new L2 signaling pathway that would explain its additional effects during cardiac ischemia. Thus, L2 is a promising candidate for drug design toward novel compounds with high potency, affinity and stability.

## 1. Introduction

Natriuretic peptide (NP)-natruretic receptor (NPR) interactions play key roles in cardiovascular homeostasis and blood pressure regulation [1,2]. The members of the mammalian NP family are atrial (ANP), B-type (BNP) and C-type (CNP) natriuretic peptides. These peptides counterbalance the renin–angiotensin system by increasing natriuresis and dieresis [1,2]. ANP and BNP are diagnostic and prognostic markers in heart failure (HF) and acute coronary syndromes (ACS), and have been also approved for the treatment of HF [3,4]. The action of NPs is mediated through their cell-surface receptors, which are a family of three homologous single-transmembrane, glycosylated receptors (NPR-A, NPR-B, and NPR-C) [5]. NPR-A and NPR-B, also called guanylyl cyclase (GC) receptors, transduce the NP signal through a large cytoplasmic domain with GC activity leading to the increase in cyclic guanosine monophosphate (cGMP) levels and the downstream activation of the protein kinase G (PKG) pathway [6]. NPR-A/B mediate most of the cardiovascular effects of NPs [7]. The third receptor, NPR-C, which does not contain a GC domain and has no direct effect on cGMP levels, serves primarily as a “clearance” receptor to remove NP hormones from the circulation [7]. In addition to its role in NP clearance, a wealth of experimental evidence indicates that NPR-C has functional significance. This receptor acts through a 37 amino acid residue cytoplasmic tail, which reportedly is phosphorylated and engages inhibitory heterotrimeric G proteins (Gi) to inhibit adenylyl cyclase (AC) and reduce subsequent cAMP levels, and to activate phospholipase C (PLCβ) [8]. Despite their different downstream signaling cascades, NPRs share a similar extracellular domain (ECD) of approximately 450 amino acid residues with ~30% homology and conserved topology, which is responsible for recognizing and binding NPs [9].

Recently, we showed that Lebetin 2 (L2), a 38-amino acid peptide (4 kDa) isolated from *Macrovipera lebetina* venom [10] and sharing structural homology with BNP [11], exerts strong cardioprotection in experimental myocardial ischemia (MI) by reducing post-ischemic necrosis, fibrosis and inflammation [12,13]. L2 cardiac actions were demonstrated to be primarily mediated through the NPR-A/cGMP signaling pathway, and alternately by NPR-B especially when NPR-A is disrupted [12]. Although the cardioprotection mechanisms of L2 and BNP were both similar and mediated by cGMP-dependent signaling, L2 exhibited additional post-ischemic actions when compared to BNP [12,13]. Only L2 increased coronary flow and endothelial cell viability, and enhanced M2-like macrophage polarization after cardiac ischemia, emphasizing potentially opposite effects of the two peptides on endothelial function [12,13].

In cardiac ischemia, myocardial injury is not limited to myocyte death but also includes coronary endothelial cell dysfunction [14]. This results in impaired coronary endothelium-dependent vasorelaxation and nitric oxide (NO) homeostasis, thereby reducing coronary perfusion and exacerbating myocardial necrosis [14]. It has been reported that NPR-C was more widely expressed in cardiac endothelial cells [15] and was able to trigger the NO signaling pathway in the vascular endothelium [16]. Therefore, a strong NPR-C-ligand binding could enhance NO signaling which in turn would restore endothelium-dependent vasodilation [17], inhibit platelet aggregation and neutrophil adhesion [18], thereby maintaining blood supply to injured myocardial cells. Since L2 induces NO-dependent signaling [12,13], it might exert its additional endothelial effects via a stronger interaction with NPR-C.

Taken together, these observations prompted us to wonder whether the discrepancies between the actions of L2 and BNP could be attributed to different affinities for NPRs, in particular NPR-C, leading to the activation of different underlying signaling pathways, hence resulting in different post-ischemic effects. Our hypothesis is further supported by studies reporting the low interaction between BNP and NPR-C [19,20], whereas venom peptides exhibit a high NPR-C binding affinity [21].

This study is intended to explore the interaction profile of L2 with NPRs which could assist experimental studies as well as the designing of L2-based therapeutics for the treatment of ischemic heart disease (IHD). A deeper understanding of the way L2 recognizes and binds to its biologically relevant receptors could lead to the understanding of biological recognition processes and underlying molecular mechanisms. In recent times, molecular docking coupled with molecular dynamics (MD) simulation studies has played critical roles in understanding the mechanism of binding interactions of potential molecules with the target proteins for lead optimization as well as design and the discovery of novel molecules [22,23]. Therefore, in this study, homology modeling, molecular docking, molecular mechanics/Poisson–Boltzmann surface area (MM/PBSA) and MD protocols have been used to investigate the binding interactions between L2 and NPRs. In silico studies were performed on mammalian (ANP, BNP, and CNP) and snake venom (L2, *dendroaspis* natriuretic peptide, DNP) NPs, and their pattern of interaction determined after binding NPRs (NPR-A, -B, and -C) in different animal species (human, rat, mouse, and chicken).

As the cardioprotective effect of L2 has been reported for the L2-alpha isoform (38 amino acids) [12,13], in this study, we used L2-alpha-Q7LZ09, the structure coordinate of which was determined by nuclear magnetic resonance (NMR) spectroscopy, and also available in the Protein Database (PDB) (PDB ID: 1Q01) [24]. In this work, we showed that L2 has a higher affinity for all human NPRs compared to BNP. As the synthetic analogues of BNP used for the treatment of HF in humans, such as nesiritide, have controversial efficacy and safety [24,25], L2 could be considered as a relevant therapeutic approach for IHD in humans. L2 also showed higher interaction with NPR-C in all species compared to BNP, and this was confirmed by both molecular docking and MD simulation studies. These differences in affinity between L2 and BNP for NPR-C could provide an explanation for the additional actions that L2 exerts during cardiac ischemia. These features make L2 a promising candidate for the development of new, more effective cardioprotection compounds with fewer adverse effects.

## 2. Results

### 2.1. Amino Acid Sequence and Model Analysis

The sequence alignment of NPs (L2, BNP, ANP CNP and DNP) and NPRs (-A, -B and -C) of human, rat, mouse and chicken was carried out in order to determine the homology between the different sequences and generate the 3D models of the proteins used in this study. NPs display high homology and identity in conserved and functional natriuretic amino acids residues and have low identity in the N-terminal and C-terminal domains (Figure 1).

L2 has the highest identity with human CNP (72.73%) and rank homology corresponding to CNP > ANP > BNP > DNP (Appendix A).

Sequence alignment of NPRs showed a high percentage of similarity in the ECD (Appendix A). A strong homology has been found between human (NPR-A, -B, and -C) and murine (rat and mouse) natriuretic receptors, and to a lesser extent with chicken receptors (Appendix A). A strong homology has also been observed between rat and mouse receptors, both sharing low homology with chicken receptors (Appendix A).

Considering the structural differences observed between the free forms of the ligands and the complex forms (Table 1) where the RMSD ranges from 13 to 19 Å between the complex form of L2 and the free form, we decided to build 3D models of all the ligands used in the work based on the 3D structure of BNP in complex with NPR-C (PDB code 1YK1) and not to use the 3D models of the ligands available in the PDB or determined by AlphaFold [26].

### 2.2. Molecular Modeling

Among the 100 3D models of each NP (Appendix A) and each receptor (Appendix A) generated by MODELLER, only one model was selected in accordance to the dope score and total energy values. The 3D model selected for NPs and NPRs was analyzed and validated in line with the recommended stereochemical quality value (Ramachandran diagram by SAVES v6.0 server, 100%) to be subsequently used for the molecular docking study.

### 2.3. Molecular Docking

The binding affinities of NPs (L2, BNP, ANP, CNP and DNP) for the natriuretic receptors (NPR-A, -B and -C) of different species, generated by the Autodock Vina 1.1.2. software, were used to compare NP–NPR interactions, within the same or between different species. 

L2 shows significantly higher affinity for human NPR-A (−25.4 ± 1.0 kcal/mol, *p* < 0.05), NPR-B (−20.6 ± 0.3 kcal/mol, *p* < 0.05), and NPR-C (−23.4 ± 0.7 kcal/mol, *p* < 0.05) compared to BNP (−18.0 ± 0.4 kcal/mol; −19.0 ± 0.8 kcal/mol; −18.3 ± 0.4 kcal/mol, respectively) (Table 2), and binds to human NPRs with a rank preference of NPR-A > NPR-C > NPR-B (−24.6 ± 1.0, −20.9 ± 0.7, −20.1 ± 0.6 kcal/mol, *p* < 0.05, respectively) (Table 3).

In all species, L2 was found to bind to NPR-C with higher affinity compared to BNP (Table 2).

Here, we only reported the results comparing L2 to BNP, used as a reference molecule for our study. However, a docking study including other NPs is reported in the Appendix A.

### 2.4. Molecular Dynamics Simulations

The stability of the receptor–ligand complexes were determined during 100 ns-long MD simulations. However, we used the data obtained after 30 ns stabilization to analyze root mean square deviation (RMSD), root mean square fluctuation (RMSF), radius of gyration (Rg), the number of hydrogen (H-) bonds formed between the ligand and the proteins and the center of mass.

#### 2.4.1. RMSD Analysis

The calculation of the RMSD provides information on the stability of the receptor–ligand complexes by comparing it to the RMSD of the receptor alone (Appendix A).

NPR-A shows a decreased deviation, thereby reflecting a better stability when it is in complex with L2 and BNP than when it is unbound. However, the complex shows significantly improved stability with L2 (2.5 ± 0.005 Å) than BNP (3.5 ± 0.007 Å, *p* < 0.001). The same pattern is found with NPR-B and NPR-C where the receptor–ligand complex is more stable when receptor is bound to L2 (4.6 ± 0.013 Å; 3.8 ± 0.006 Å) compared to BNP (6.0 ± 0.008 Å, *p* < 0.001; 4.1 ± 0.014 Å, *p* < 0.001).

#### 2.4.2. RMSF Analysis

The RMSF plot (Appendix A) shows that at the level of the receptor–L2 complex, a decrease in fluctuation was observed to stabilize in an average value of 1.1 ± 0.013 Å for NPR-A and 1.2 ± 0.012 Å for NPR-C, while the BNP fluctuation was observed in an average value of 1.3 ± 0.015 Å for NPR-A (*p* < 0.001) and 1.5 ± 0.018 Å for NPR-C (*p* < 0.001). However, there is no significant difference in the stability of the receptor–ligand complex when both molecules bind to NPR-B (L2: 1.9 ± 0.020 Å vs. BNP: 1.8 ± 0.023 Å).

#### 2.4.3. Radius of Gyration Analysis

The computation of the Rg gives a global view of the compactness of a receptor with its ligand. Rg plots (Appendix A) show that NPR-A, -B and -C have a better stability when they bind to L2 (29.8 ± 0.003 Å; 31.3 ± 0.008 Å; 30.8 ± 0.004 Å) than BNP (30.5 ± 0.003 Å, *p* < 0.001; 30.5 ± 0.003 Å, *p* < 0.001; 31.4 ± 0.009 Å, *p* < 0.001).

#### 2.4.4. Hydrogen Bonds Analysis

The number of hydrogen bonds (H bonds) formed throughout the simulation plays a key role in the formation and stability of the receptor–ligand complex. 

A first analysis at 3 Å distance between receptor and ligand was performed to compare the number of H bonds formed in NPR–L2 and NPR–BNP complexes (Appendix A). A significant increase in H bonds following the binding of L2 to NPR-A (7.1 ± 0.062), -B (10.7 ± 0.073) and -C (7.4 ± 0.063) was observed compared to BNP (6.8 ± 0.060, *p* < 0.001; 1.5 ± 0.038, *p* < 0.001; 6.2 ± 0.059, *p* < 0.001, respectively). The analysis of the number of H bonds developing the best interactions between receptor and ligand (at 3 Å) indicated that L2 has a higher number compared to BNP for NPR-A (L2: 14, BNP: 10) and -B (L2: 16, BNP: 11). For NPR-C, there is no difference in the number of H Bonds between L2 and BNP (12).

Using PyMol, residues involved in the H-bond interactions allowing the best receptor–ligand stability were specifically determined throughout the simulation (Figure 2 and Table 4).

A second analysis at 4.5 Å showed that L2 has an increased number of H Bonds for NPR-A (607), -B (791) and -C (671) compared to BNP (496, 605 and 611, respectively).

#### 2.4.5. Center of Mass Analysis

The center of mass is calculated to determine the distance separating the center of mass of the ligand from the center of mass of the receptor during the simulation to determine the behavior of the ligand throughout the simulation. 

For L2, no detachment from NPR-A, was observed throughout the simulation with a basal value of 3.8 Å, and a value of 4.3 Å at the end of the simulation (Appendix A).

However, under the same simulation conditions, BNP revealed instability to NPR-A, due to its detachment throughout this study, with values ranging from 12.0 Å at 30 ns to 16.8 Å at 70 ns. This result was evidenced by the mean values of L2 (4.7 ± 0.016 Å) and BNP (11.4 ± 0.139 Å, *p* < 0.001). Regarding NPR-B, L2 showed instability (values ranging from 9.9 Å at 30 ns to 22.4 Å at 70 ns, mean value: 10.5 ± 0.187 Å), while with BNP no detachment was observed (values ranging from 8.9 Å at 30 ns to 7.5 Å at 70 ns, mean value: 7.9 ± 0.017 Å, *p* < 0.001). None of the complexes of NPR-C with L2 or BNP showed optimal stability, with values ranging from 13.5 Å at 30 ns to 19.2 Å at 70 ns for L2, and from 9.4 Å at 30 ns to 14.2 Å at 70 ns for BNP.

#### 2.4.6. Binding Free Energy Analysis by the MM/PBSA Method

To determine the affinity of L2 and its efficiency in binding natriuretic receptors, we have calculated the relative binding energy MM-PBSA, which is one of the most reliable and widespread approaches combining the models of molecular mechanics and continuous solvent to calculate G ΔG°–bind of small molecules. By comparing the total relative binding energy of the NPR-2 complex, which is −246.2 ± 6.2, −248.0 ± 10.6, and −232.4 ± 9.5 kcal/mol for NPR-A, -B and -C, respectively, to the relative binding energy presented by the NPR–BNP complex, which is −249.3 ± 5.7, −255.5 ± 8.5 and −200.8 ± 8.8 kcal/mol for the three receptors, we can conclude that L2 has a significantly higher interaction with NPR-C than that observed with BNP (*p* < 0.05) under the same working conditions (Appendix A).

#### 2.4.7. Receptor–Ligand Interaction Analysis

The electrostatic and hydrophobic interactions were evaluated firstly at 4.5 Å by Protein Structure and Interaction Analyzer (PSAIA), and afterwards at less than 2.5 Å by the LigPlot+ 2D receptor–ligand interaction diagrams, to provide information on the stability of the receptor–ligand complexes.

PSAIA study (Table 5 and Table 6) demonstrated that L2 revealed a balanced interaction with the three human receptors (NPR-A, -B and -C) since the residues involved in the NPR–L2 interaction are well distributed between the two chains of receptors. However, unlike L2, BNP displays unbalanced interactions with all three NPRs, and in particular NPR-C, which could indicate a lower affinity of BNP compared to L2 for natriuretic receptors.

We found that L2 has more aromatic π-type interactions for the three receptors compared to BNP, which confirm a better interaction of L2 with human NPRs (Table 5 and Table 6).

L2 interacts at its Phe15 residue with NPR-A at Phe165 residue in the A chain and Tyr154 and Tyr165 residues in the B chain, while no aromatic interaction was found for BNP with NPR-A.

For NPR-B, L2 interacts at its Phe15 residue with Phe166 residue in the A chain and Tyr181 residue in the C chain, while no aromatic interaction was found for BNP with NPR-B.

For NPR-C, L2 interacts at its Phe15 residue with Phe169 residue in the A chain and Tyr181 and Phe190 residues in the B chain, while only one aromatic interaction was found for BNP interacting at its Phe11 residue with NPR-C at Tyr347 in the A chain.

LigPlot+ 2D diagrams (Appendix A) showed that L2 has higher total number of interactions compared to BNP with 16 and 18 interactions for NPR-A and -C, respectively (Appendix A), while the number of interactions of BNP to these receptors were 14 and 13, respectively (Appendix A). However, for NPR-B, we have found the same number of interactions (20) for both L2 and BNP (Appendix A). This confirms the stronger electrostatic interaction of L2 with NPR-A and NPR-C.

Furthermore, LigPlot+ study confirmed the presence of conserved and functional natriuretic amino acids residues for L2 and BNP. Analysis of the electrostatic interactions between receptor and ligand indicates that for NPR-A and NPR-B, a single basic amino acid of L2 or BNP interacts with an acidic amino acid of the receptor. Regarding NPR-C, L2 and BNP have the same mode of interaction, using two basic amino acids from the ligand that interact with one acidic amino acid from the receptor (For details, see Appendix A).

## 3. Discussion

In the present study, we performed various in silico approaches to investigate the molecular interaction, binding mode and major interaction residues in order to provide new mechanistic insights into the interaction of L2 with natriuretic receptors, and more broadly into the mechanism of action of NPs.

To achieve this goal, we performed molecular docking and MD simulation analysis on L2 and its interaction with NPRs (NPR-A, -B, and -C). Since we are targeting cardioprotection in humans, most work has been focused on L2 and human NPRs. L2 has been compared mainly to BNP as a reference molecule, as the two peptides show structural homology towards cardiovascular properties and have been compared previously for their cardiovascular effects [12,13]. Furthermore, we investigated the interaction of L2 with other NPRs species (rat, mouse, and chicken), and compared obtained results to those regarding other NPs (ANP, CNP, and DNP) (Appendix A).

The main data show that, compared to BNP, L2 binds with higher affinity to human NPRs, with a rank receptor preference of NPR-A > NPR-C > NPR-B. Interestingly, all studies, i.e., molecular docking, MD simulation and receptor–ligand interaction analysis, demonstrated the superiority of L2 binding to NPR-C compared to BNP, and this has particularly been observed in all species (Table 2).

Our results are in accordance with previous in silico studies reported by Mosbah et al. which demonstrated that the sequence alignment and the 3D structure of L2 have high similarities to NPs and suggested that L2 may act as a natriuretic factor [24]. This was additionally supported in experimental studies which proved that L2 utilizes identical mechanism for cardioprotection as BNP by activating NPRs and subsequently mitochondrial KATP channels, and inhibiting mitochondria-induced apoptosis [12]. More specifically, L2 was observed to act through NPR-A/cGMP-mediated signaling [12], which is in line with our current in silico data highlighting the role of this receptor in L2-mediated effects. The extensive analysis of the electrostatic interactions of L2 and BNP with the three natriuretic receptors (NPR-A, -B and -C) (Appendix A) confirms the high affinities of two ligands with these receptors. Albeit L2 and BNP exert cardioprotection through the same NPR-A signaling pathway, L2 was found to have additional cardiac effects compared to BNP [12,13]. In this work, we found that L2 has more aromatic π-type interactions with NPR-A, which are particularly strong bonds allowing a stable receptor–ligand interaction (Table 5 and Table 6). In this context, three aromatic interactions were found for L2 and NPR-A while none was found for BNP with the same receptor (Table 5 and Table 6).

Only L2 increased coronary flow, enhanced endothelial cell viability, improved severe cardiac dysfunction and improved the resolution of post-ischemic inflammation [12,13]. As these observations converge towards a better improvement of endothelial function [13], we suggested that L2 could exert these potential endothelial actions by potentiating a NO endothelium-dependent pathway possibly via NPR-C. The latter may play a role in NO signaling at the vascular endothelium, since its stimulation induces a G protein-dependent activation of smooth muscle endothelial NO synthase (eNOS) [25] and phosphorylation of the PI3K/Akt/NO pathway [16]. Unlike NPR-A/B, NPR-C is mainly expressed in endothelial and vascular smooth muscle cells [27] where its density is highest compared to all other receptors (94% of the total population of NPRs) [15]. In addition to being admitted as a clearance receptor, NPR-C was demonstrated to mediate specifically physiological effects of NPs in the heart and vasculature. Substantial body of biochemical work has demonstrated that these novel actions are promoted by the ability of NPR-C to couple Gi proteins and cause downstream inhibition of AC and activation of PLC [8]. Other reports showed that the activation of NPR-C can lead to a selective inhibition of L-type calcium currents in cardiomyocytes and to an endothelium-derived hyperpolarization which regulates local blood flow in the coronary arteries and systemic blood pressure by hyperpolarizing smooth muscle cells [28]. Hence, due to its inhibitory effects on L-type calcium currents in cardiomyocytes, improvement of endothelial-dependent vasorelaxation in coronary arteries, and stimulation of PLC-mediated signaling, NPR-C activation might be of particular interest in the setting of cardiac ischemia. Here, we found that NPR-C binds L2 to a higher extent than BNP in all species (Table 2) suggesting that L2 may exert its potential endothelial effects via a stronger interaction with this receptor. Overall, by referring to electrostatic and hydrophobic aromatic interactions, our data suggest docking or guided binding of L2 with all three NPRs (NPR-A, -B, and -C), especially for NPR-A and NPR-C (Table 5 and Table 6). Our results are supported by other in silico studies showing structural similarity between L2 and peptides with strong binding for NPR-C. Specifically, the primary sequence of L2 shows similarity to the C-terminal fragment of osteocrine, a natriuretic-like peptide with a strong interaction with NPR-C [29]. Pharmacological studies have also indicated that, like L2, other snake venom-derived NPs such as DNP display a strong interaction profile with NPR-C [21]. More interestingly, L2 induces phosphorylation of the PI3K/Akt/NO signaling pathway in isolated ischemic rat hearts [12]. The cross-checking of the current computational study with experimental data obtained in animals is in agreement with a major potential role of NPR-C in the cardioprotective effects of L2. It is well established that BNP binds to NPR-C with low affinity compared to other NPs [19,20], which makes its availability in the blood longer (half-life of approximately 22 min compared to 3 min for ANP [3,30,31]) and justifies its use as a clinical diagnostic biomarker in coronary heart disease. Having confirmed in silico (Table 2, Table 3, Table 5 and Table 6) these results, it is thus possible that the low BNP–NPR-C interaction may explain why BNP exerts less cardiac actions than L2. However, more experimental and pharmacological investigations are needed to confirm these in silico results and to further explore the direct role of NPR-C in the cardiac effects of L2 and BNP, as well as the role of endothelial function in the observed effects.

In silico data should be extrapolated with caution to clinical and preclinical settings. However, several arguments allow the validation of the current work and therefore the extrapolation of the in silico predictive data found on L2 to physiological situations. For instance, we have determined and confirmed for the BNP the same conserved and functional natriuretic amino acids residues (Appendix A) as those reported in the literature [32]. Moreover, we have demonstrated that venom-derived DNP binds to human NPR-A, NPR-C and NPR-B, with decreasing order of affinity (Appendix A). Similar findings have been previously reported by pharmacological studies showing that DNP has a high pM affinity for human NPR-A, nM affinity for NPR-C and no affinity for NPR-B (Ki > 1000 nM) [21]. Consistent with DNP data, L2 presents the same binding pattern to NPRs (NPR-A > NPR-C > NPR-B) in our study. For the mammalian NPs, as expected [20], ANP binds to NPRs with an affinity preference to NPR-A in humans and other species (Appendix A). ANP binding to NPRs was also found stronger than with BNP, and this has been observed regardless of receptor type or species (Appendix A). Consistent results were also found with CNP which binds with an affinity preference to NPR-B in all species, and with a rank preference in human receptors of NPR-B > NPR-C > NPR-A, (Appendix A), thus confirming the almost exclusive specificity of CNP for NPR-B [28]. Regarding BNP, we found higher affinity (Table 3) and dynamic interaction (Appendix A) to NPR-B, whereas in the literature, NPR-A activation remains the predominant mechanism mediating BNP actions [20]. Moreover, even if according to the analysis of receptor–ligand interactions (Table 4 and Table 6), BNP binds to NPR-A and NPR-B, this interaction would not be effective due to an imbalance of hydrophobic residues of BNP interacting with the two chains of receptors. Further experiments are needed to better understand the role of NPR-B in the effects of BNP and to validate our current in silico data. These receptors, although expressed in the heart with lower abundance than NPR-A [33], are reported to be the most active NPRs in the failing heart [34]. They can be overactivated upon BNP overexpression [35] and lead to the same action as NPR-A binding [36,37] especially when NPR-A is disrupted [12].

## 4. Conclusions

The present study shows that L2 potently binds to NPRs with a strong interaction to human receptors. Moreover, L2 was found to interact more potently with NPR-A than BNP and to have higher binding affinity to NPR-C in all species. Our findings provide innovative insights into the molecular mechanism of action of L2 in cardioprotection and myocardial repair. L2 would simultaneously activate two signaling pathways, one triggering cardiomyocyte repair (via NPR-A) and the other leading to endothelial cell repair and endothelium-dependent vasorelaxation (via NPR-C), whereas BNP, due to its weak binding to NPR-C, would induce only one pathway (NPR-A). Since cardiac ischemia does not only involve cardiomyocyte injury but also endothelial damage, L2 treatment would have a dual action on the two pathways during MI. These observations are in line with previous experimental work [12,13] supporting that L2 could exert additional protective effects during MI through endothelial function improvement and NPR-C activation in post-MI. Synthetic BNP analogues such as nesiritide are used in acute decompensated HF in humans, but issues have been raised in relation to their safety and effectiveness [38,39]. Hence, clinical trials have focused on examining new peptides, such as the chimeric peptide CD-NP (cenderitide) designed from the venom peptide DNP, but clinical studies have been halted [40,41]. In this context, L2 would be a promising candidate for drug design towards new compounds with high activity, affinity, stability, and fewer adverse effects and immunogenicity issues. These features also pave the way to the chemical synthesis of L2 analogues, which is currently in progress to develop clinically relevant therapeutic molecules for the treatment of IHD.

## 5. Methods

In the current study, the design of the 3D models, docking and molecular simulation studies were performed in silico using the Exxact System CentOs Linux 7 bioinformatics modeling workstation (Exxact Valence VWS-1542881-AMB).

### 5.1. Receptor–Ligand Preparation

#### 5.1.1. Multiple Sequence Alignment

The amino acid sequence of NPRs from different species (human, rat, mouse, and chicken) as well as those of NPs (ANP, BNP, CNP, DNP and L2) were collected under the FASTA format from the UniProtKB database [42]. Clustal Omega (http://www.clustal.org) (accessed on 12 December 2019) as used for multiple alignments to determine the percentages of identities and coverages between NPRs of different species and between the NPs.

#### 5.1.2. Receptor Structure Preparation

The ECD of NPRs from different species (human, rat, mouse, and chicken), unbound or bound with NPs, was extracted from its crystallographic structure coordinates available in the Protein Database (PDB) [43] (https://www.rcsb.org) (accessed on 7 December 2019) with access codes PDB ID: 1T34 [44] for the *Rattus norvegicus* NPR-A–ANP complex; PDB ID: 1DP4 [45] for *Rattus norvegicus* free NPR-A receptor; PDB ID: 1YK0 [32] for the NPR-C–ANP complex of *Homo sapiens*; PDB ID: 1YK1 [32] for the NPR-C–BNP complex of *Homo sapiens;* and PDB ID: 1JDP [46] for the NPR-C–CNP complex of *Homo sapiens*.

#### 5.1.3. Ligand Structure Preparation

Mammalian- (ANP, BNP, and CNP) and snake venom-derived (DNP, L2) NPs were used to study and compare their interactions with NPRs (NPR-A, -B, and -C) in different species. The x-ray crystallographic structure of the receptor–ligand complexes was downloaded from the PDB to obtain the structure coordinates of NPs, except for L2, the coordinate structure of which was determined by NMR spectroscopy also available in the PDB (PDB ID: 1Q01) [24]. The sequences of NPs were collected under the FASTA format from the UniProtKB database (ANP ID: P01160; BNP ID: P16860; CNP ID: P23582; DNP ID: Q8QGP7; and L2 ID: Q7LZ09) in order to build their 3D models by MODELLER 9.23 (San Francisco, CA, USA).

#### 5.1.4. Molecular Modeling

Molecular modeling was carried out in four steps. First, a blast homology search was performed using the program BlastP (Protein Basic Local Alignment Search Tool, Bethesda, MD, USA) [47] to identify the templates for each receptor and ligand according to their percentage identity, the coverage of the sequence as well as the E-value. Second, the template and target sequences were aligned using the “EMBOSS Needle” program [48]. Third, the models were built using a python script, the functionality of which is implemented in the MODELLER 9.23 program [49]. One hundred conformers for each model were generated and the lowest Discrete Optimized Protein Energy (DOPE) score model was selected for each. Fourth, SAVES v6.0 [Protein Structure Analysis and Verification Server (https://saves.mbi.ucla.edu/ accessed on 11 July 2020)], was applied to validate the 3D models of receptors (NPRs) and ligands (NPs) by verifying their Ramachandran Plot. In addition, NAMD version 2.13 software [50,51] was used to perform energy minimization and structure refinement of the modeled proteins using the CHARMM force field. Structural visualization was performed in PyMOL (http//www.pymol.org) (accessed on 9 November 2019) [52] and the VMD (Visual Molecular Dynamics) program [50,51].

#### 5.1.5. Receptor–Ligand Molecular Docking

Molecular docking algorithms are used in structural bioinformatics to predict the 3D structure of protein–protein complexes from the unbound structures of a ligand with all rotatable bonds free and a receptor with all bonds rigid. The ligand structure (NP) was docked to the receptor structure (NPR) using Autodock Vina 1.1.2. software embedded in MGL Tools version 1.5.6 software [53] in order to determine the stability and the binding energy of each complex. During the docking procedure, L2, ANP, BNP, CNP and DNP were used as ligands, with almost all of their bonds defined as no rotatable. All receptors were kept rigid. Grid maps representing the target proteins were constructed with different dimensions depending on the active site of the target protein. Autodock Tools 1.1.2. software was used to prepare ligand and receptor files (pdbqt), attribute charges and calculate grid box. Based on the Autodock Tools software and during the preparation of the receptors, the protonation was carried out by adding only polar hydrogens, and applying the Kollman charges for the proteins, this is available in the “Edit” menu of the Autodock Tools software. For the ligands the Gasteiger charges were applied and all these treatments were carried out on the basis of the physiological pH. During docking, the parameters corresponding to the genetic-algorithm were set by default with ga_pop_size 150 (number of individuals in the population) and ga_num_evals 25,000,000 (maximum number of energy evaluations). The best complexes were selected and analyzed according to their most favorable Vina scores expressed as Gibbs free binding energies (∆G°) in kcal/mol unit [54,55].

### 5.2. Molecular Dynamics Simulation

The most powerful method used for MD simulation is NAMD v.2.13 software, which works with the AMBER potential and based on the polarizable force field CHARMM36 [50]. First, the NPR–NP complex structures were minimized and equilibrated using MD for 20,000 cycles. Then, each complex was solvated in a cubic box containing water molecules. Sodium and chloride ions (255 mM and 9 mM) were added to neutralize the whole box. Subsequently, the charged system was harmonized for 60,000 cycles. Finally, all the harmonized complexes were conducted under normal conditions for temperature and pressure (NPT conditions) for 100 ns dynamic stimulation step at 310 K room temperature and 1 bar atmospheric pressure, using a time step of ∆t = 2 fs with the cutoff distance 12.0 Å, using Langevin dynamics with a damping constant of 1 ps-1. Energy minimization was performed for 1000 steps. After the minimization step, the last frames were used to start MD simulations for 130 ps using CHARMM36/NAMD. Only the complex which presents the strongest potential will be simulated in MD simulation for 100 ns [51,56]. The root mean square deviation (RMSD) and root mean square fluctuation (RMSF) were calculated using the VMD tools (Urbana-Champaign, Champaign, IL, USA) [57,58]. The atomic charges of drugs studied in the dynamics simulation were assigned by adding parameter files in the dynamic script such as ligand.prm generated from charmm-gui (corresponding to ligand) and other parameter files such as par_all36_prot.prm, par_all36_na.prm, par_all36_lipid.prm, par_all36_cgenff.prm, par_all36_carb.prm, par_all27_prot_lipid.inp, par_all22_prot.prm. Complete trajectories and analysis of receptor–ligand interactions were visualized in VMD

The complete trajectories and the analysis of receptor–ligand interactions were visualized in VMD.

#### 5.2.1. Analysis of RMSD

The root mean square deviation (RMSD) was evaluated by a quantitative method using NAMD v.2.13 software, the CHARMM36 force field and VMD program to compare the stability of two molecules during a 100 ns MD simulation.

#### 5.2.2. Analysis of RMSF

Residual root mean square fluctuation (RMSF) was obtained by NAMD v.2.13 /CHARMM36 software and was analyzed by VMD program to identify the different spatial fluctuations of each individual residue in bound and unbound protein structure during the 100 ns MD simulation. RMSF values provided information on structural flexibility, thermal stability and heterogeneity of macromolecules.

#### 5.2.3. Analysis of Radius of Gyration

The radius of gyration (Rg) was evaluated by NAMD software v.2.13 using CHARMM36 and VMD program. Rg values provide information on the protein–protein dimension to determine the stability and configurational compactness in each complex.

#### 5.2.4. Dynamics of Hydrogen Bonds

The number of hydrogen bond (H bonds) interactions was calculated using H bond occupancy function in VMD program in order to investigate the conformational changes and the stability of the NPR-NP complexes. In our analysis, we selected the major number of H bonds by frame generated at 3 Å and 4.5 Å. The most important H-bond interactions at 3 Å between the receptor and the ligand were visualized by PyMol 2.4.0. software and the residues involved in the interactions identified.

#### 5.2.5. Analysis of the Center of Mass

The center of mass is a specific point to determine the movements between two proteins at different linear, kinetic and dynamic instants. The distances between the centers of mass of receptor (NPR-A, -B, and -C) and ligand (L2 and BNP) were measured using NAMD v.2.13/CHARMM36 software and were analyzed by VMD program according to the initial position during the 100 ns MD simulations.

#### 5.2.6. Analysis of Binding Free Energy

The binding free energy of the complexes NPR–L2 and NPR–BNP was determined using the molecular mechanics Poisson–Boltzmann solvent accessible surface area (MM-PBSA) method [59]. The total binding free energy (∆Gbinding for the receptor–ligand complexes were calculated with a total of 10 snapshots extracted from the trajectories between 30 and 100 ns simulation time (when systems are stabilized) with a gate spacing fixed at 0.5 Å in accordance with the Equation (1) below [60]:∆G_bind_ = G_complex_ − (G_receptor_ + G_ligand_)(1)
where G_complex_, G_recepror_, and G_ligand_ represent the free energies of complex, receptor, and ligand averaged over random snapshots taken from MD.
∆G_bind_ = ∆E_MM_ + ∆G_solv_ − T∆Swhere(2)
∆E_MM_ = ∆E_internal_ + ∆E_electrostatic_ + ∆E_vdw_(3)
∆E_internal_ = ∆E_bond_ + ∆E_angle_ − ∆E_torsion_(4)
∆G_solv_ = ∆G_polar_ + ∆G_non-polar_(5)

Finally, the MM-G/PBSA value of the receptor–ligand complex was calculated from the gas-phase MM energy (∆E_MM_), solvation energy (∆G_solv_) and the entropic term (−T∆S) that it also involves the summing of the internal energies ∆E_internal_ [bond (∆E_bond_), angle (∆E_angle_) and dihedral torsion (∆E_torsion_) energies], electrostatic energy (∆E_electrostatic_) and van der Waals energies (∆E_vdw_), polar (∆G_polar_), non-polar (∆G_non-polar_) components, and the entropic term (−T∆S).

The average mean values of RMSD, RMSF, Rg, H bonds, center of mass and binding free energy were used to evaluate the stability of NPR–NP complexes after stabilization of 30 ns of MD simulation. The lower value of RMSD, RMSF, Rg, center of mass and binding free energy indicates a more stable NPR–NP complex. For H bonds, the higher value indicates a more stable receptor–ligand complex.

#### 5.2.7. Receptor–Ligand Interaction Analysis

To further study and compare the strength of receptor–ligand binding, a visual analysis of the complex structure was performed by PyMOL (http//www.pymol.org) (accessed on 6 November 2019) and VMD program. This approach is based on the output files of MD simulations to analyze the pockets of interaction between the receptor and the ligand during 100 ns-long MD simulations. It is used to monitor intermolecular H bonds and electrostatic and hydrophobic interactions in NPR–NP complexes. The 2D and 3D structural images of the interaction between the receptor and the ligand were generated and visualized by PSAIA [61] (Protein Structure and Interaction Analyzer) at 4.5 Å, and LigPlot+ [62] at 2.6 Å, to predict protein–protein interaction sites.

#### 5.2.8. Statistical Analysis

To establish a linear correlation between the binding affinities and the different Gibbs free energy (ΔG°) data emitted by the receptor–ligand complexes, a correlation coefficient between the values of ΔG° was calculated. Statistical analysis between groups was performed using ANOVA. Values of *p* < 0.05 were considered statistically significant.

## Figures and Tables

**Figure 1 toxins-14-00787-f001:**
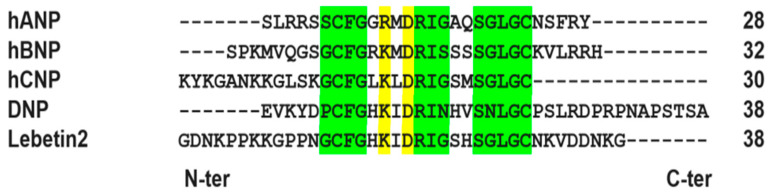
Multiple alignments of the amino acid sequences of Lebetin-2-alpha (Q7LZ09) (L2) compared to amino acid sequences of human natriuretic peptides (NPs): atrial natriuretic peptide (ANP), B-type natriuretic peptide (BNP) and C-type natriuretic peptide (CNP), and snake venom-derived *dendroaspis* natriuretic peptide (DNP). The referenced functional natriuretic amino acids residues are dashed in green. The conserved amino acids in all natriuretic peptides are dashed in green and yellow. All natriuretic peptides have a carboxylic C-termini and amidated N-termini. The numbers in the last column correspond to the length of the peptides.

**Figure 2 toxins-14-00787-f002:**
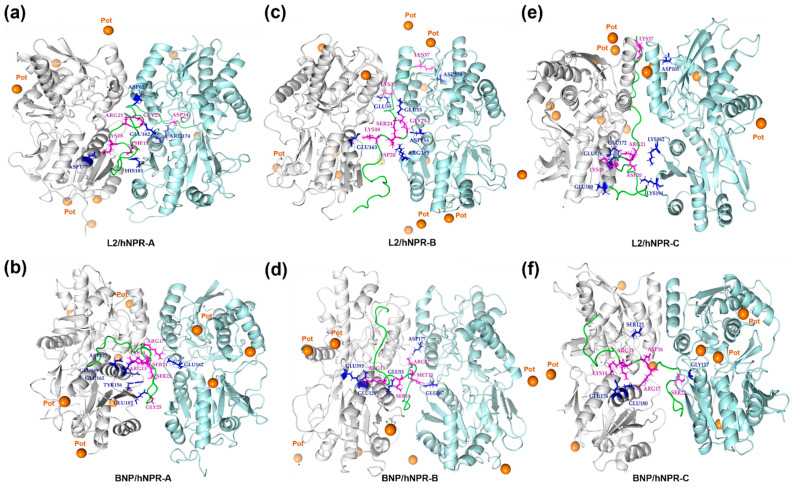
PyMol representation of hydrogen interactions detected between natriuretic receptors (NPRs) and L2 (**a**,**c**,**e**) and BNP (**b**,**d**,**f**). NPR dimers, L2 and BNP are represented in cartoons. Below, A, B and C are the chains of the natriuretic receptors (A and B are the chains of NPR-A and NPR-C and A and C are the chains of NPR-B). (**a**) The amino acids of L2, Phe15, Lys18, Arg21, Gly23, and Asp34 form hydrogen bonds with NPR-A His185 (B), Asp177 (A), Asp62 (B), Glu162 (B) and Arg175 (B), respectively. (**b**) The amino acids of BNP, Arg13, Lys14, Arg17, Ser21, Ser22, and Gly25 form hydrogen bonds with NPR-A Tyr156 (A), Glu162 (A), Glu 169 (A), Asp77(A), and Glu162 (B), respectively. (**c**) The amino acids of L2, Lys18, Asp20, Gly23, Ser24, LYS32, and Lys37 form hydrogen bonds with NPR-B Glu163 (A), Arg183 (C), Asp155 (C), Glu55 (C), Glu55 (A), and Asp338 (C), respectively. (**d**) The amino acids of BNP, Arg13, Met15, Arg17, and Ser19 form hydrogen bonds with NPR-B Glu129 (A), Glu167 (A), Asp155 (C) and Glu55 (C), respectively. (**e**) The amino acids of L2, Lys18, Asp20, Arg21, and Lys37 form hydrogen bonds with NPR-C Glu176 (A), Lys194 (B), Glu176 (A), and Asp105 (B), respectively. (**f**) The amino acids of BNP, Arg13, Lys14, Asp16, Arg17, and Ser22 form hydrogen bonds with NPR-C Glu180 (A), Glu1176 (A), Arg99 (A), Glu176 (A), and Gly117 (B), respectively. Lebetin 2 (L2) and human B-type natriuretic peptide (BNP) are represented in green. The potassium ion (K^+^) are in orange, NPR monomer 1 in cyan blue and monomer in gray. L2 and BNP residues in purple are involved in interaction with NPR residues (in blue).

**Table 1 toxins-14-00787-t001:** RMSD of the free L2 MODELLER model compared to 20 NMR models (PDB code 1Q01) and the free L2 AlphaFold model (Q7LZ09).

RMSD (Å)	L2 MODELLER Model
AlphaFold_L2 model	16.576
1Q01_01 model	14.110
1Q01_02 model	18.277
1Q01_03 model	14.878
1Q01_04 model	17.960
1Q01_05 model	14.591
1Q01_06 model	16.407
1Q01_07 model	14.373
1Q01_08 model	17.637
1Q01_09 model	17.412
1Q01_10 model	16.558
1Q01_11 model	13.952
1Q01_12 model	16.115
1Q01_13 model	13.961
1Q01_14 model	14.061
1Q01_15 model	18.362
1Q01_16 model	13.592
1Q01_17 model	16.661
1Q01_18 model	19.069
1Q01_19 model	16.237
1Q01_20 model	14.835

L2, Lebetin 2; NMR, nuclear magnetic resonance; RMSD; root mean square deviation.

**Table 2 toxins-14-00787-t002:** Inter-species comparative study of the binding affinity of L2 and BNP to natriuretic receptors.

	Human NPRs		Rat NPRs		Mouse NPRs		Chicken NPRs	
NPR-A	L2	BNP	L2	BNP	BNP	L2	L2	BNP
ΔG° (kcal.mol)	−25.4 *	−18.0	−24.0 *	−16.9	−22.5	−19.9	−19.0 *	−15.4
SEM	1.0	0.4	0.8	0.2	0.2	0.8	0.2	0.2
NPR-B	L2	BNP	L2	BNP	L2	BNP	L2	BNP
ΔG° (kcal.mol)	−20.6 *^,†^	−19.0 ^†^	−24.0	−17.0	−23.9	−22.4	−23.1 *	−18.4
SEM	0.3	0.8	0.4	0.3	0.4	0.2	0.1	0.1
NPR-C	L2	BNP	L2	BNP	L2	BNP	L2	BNP
ΔG° (kcal.mol)	−23.4 *	−18.3 ^£^	−15.5 *	−14.4	−23.0 *	−19.5	−20.1 *	−17.7
SEM	0.7	0.4	0.2	0.1	0.2	0.1	0.2	0.2

Data were generated by a molecular docking study using Autodock Vina 1.1.2. software. Natriuretic peptides (NPs) in each row were ranked in decreasing order of interaction with the corresponding natriuretic receptor (NPR) (i.e., the lower the interaction energy, the better the stability of the receptor–ligand complex). BNP, B-type natriuretic peptide; L2, Lebetin 2; NPR, natriuretic peptide receptor (NPR); NPR-A, natriuretic peptide receptor A; NPR-B, natriuretic peptide receptor B; NPR-C, natriuretic peptide receptor C. ΔG°, Gibbs free energy. *N* = 3 values/condition.*, *p* < 0.05 vs. corresponding BNP; †, *p* < 0.05 vs. corresponding NPR-A; £, *p* < 0.05 vs. corresponding NPR-B.

**Table 3 toxins-14-00787-t003:** Intra-species comparative study of the binding affinity of L2 and BNP to natriuretic receptors.

	L2			BNP		
Human	hNPR-A	hNPR-C	hNPR-B	hNPR-B	hNPR-A	hNPR-C
ΔG° (kcal.mol)	−24.6	−20.9 ^†^	−20.1 ^†^	−16.2	−15.8	−14.1
SEM	1.0	0.7	0.6	0.4	0.5	0.1
Rat	rNPR-A	rNPR-B	rNPR-C	rNPR-B	rNPR-A	rNPR-C
ΔG° (kcal.mol)	−23.3	−17.04 ^†^	−17.0 ^†^	−16.0	−14.9 ^£^	−13.6 ^£^
SEM	1.0	0.4	0.3	0.2	0.5	0.0
Mouse	mNPR-B	mNPR-C	mNPR-A	mNPR-A	mNPR-B	mNPR-C
ΔG° (kcal.mol)	−18.9	−18.5	−17.9	−18.3	−15.8	−13.5 ^†^
SEM	0.4	0.5	0.4	1.0	0.2	0.1
Chicken	chNPR-B	chNPR-A	chNPR-C	chNPR-B	chNPR-A	chNPR-C
ΔG° (kcal.mol)	−21.5	−20.1	−19.0 ^£^	−15.0	−14.7	−14.4
SEM	0.4	0.4	0.4	0.2	0.5	0.4

Data were generated by a molecular docking study using Autodock Vina 1.1.2. software. The natriuretic peptide receptors (NPR) of each line were ranked in decreasing order of affinity with the corresponding natriuretic peptide (L2 and BNP), i.e., the lower the interaction energy between the receptor and the ligand, the better the stability of the complex. BNP, B-type natriuretic peptide; L2, Lebetin 2; chNPR, chicken NPR; hNPR, human NPR; mNPR, mouse NPR; rNPR, rat NPR. ΔG°, Gibbs free energy. *N* = 3 values/condition. †, *p* < 0.05 vs. corresponding NPR-A; £, *p* < 0.05 vs. corresponding NPR-B.

**Table 4 toxins-14-00787-t004:** Residues involved in the hydrogen bond interactions between natriuretic receptors and their ligands (L2 and BNP).

	hNPR-A	hNPR-B	hNPR-C
L2	**GLY1**-GLU119 (A) **LYS4**-GLU162 (B) **LYS7**-ASP192 (B) **LYS7**-ASP191(B) **LYS8**-GLU187 (B) **GLY13**-GLU187 (B) **PHE15**-HSD185 (B) **LYS18**-ASP177 (A) **ARG21**-GLU162 (A) **ARG21**-GLU162 (B) **GLY23**-GLU162 (B) **ASP34**-LYS132 (B) **ASP34**-ARG174 (B) **ASP34**-ARG178 (B)	**GLY1**-PHE419 (A) **ASP2**-ARG143 (A) **ASP2**-ARG200 (A) **LYS7**-GLU189 (C) **PRO11**-GLN177 (A) **GLY13**-GLU184 (C) **LYS18**-GLU163 (A) **ASP20**-ARG183 (C) **ARG21**-GLU55 (C) **GLY23**-ASP155 (C) **SER24**-GLU55 (C) **LYS32**-GLU55 (A) **ASN36**-HSD92 (C) **LYS37**-GLU337 (C) **LYS37**-ASP338 (C) **GLY38**-ARG1 (C)	**GLY1**-GLU180 (A) **GLY1**-GLU181 (A) **LYS4**-GLU180 (A) **LYS7**-ASP197 (B) **LYS7**-ASP200 (B) **LYS18**-GLU176 (A) **LYS18**-GLU180 (A) **ASP20**-LYS162 (B) **ASP20**-LYS194 (B) **ARG21**-GLU172 (A) **ARG21**-GLU176 (A) **LYS37**-ASP105 (B)
BNP	**LYS3**-ARG178 (A) **ARG13**-ASP177 (A) **LYS14**-GLU169 (A) **ARG17**-GLU162 (B) **SER21**-GLU162 (A) **SER22**-TYR156 (A) **SER22**-GLU169 (B) **GLY25**-GLU187 (A) **LYS27**-GLU187 (A) **ARG31**-GLU169 (B)	**SER1**-THR116 (A) **SER1**-GLN347 (A) **ARG13**-GLU129 (A) **ARG13**-GLU393 (A) **MET15**-GLU167 (A) **ARG17**-ASP155 (C) **SER19**-GLU55 (C) **SER22**-TYR56 (A) **SER21**-ARG88 (C) **ARG30**-GLU184 (C) **ARG31**-ASP155 (C)	**ARG13**-GLU180 (A) **LYS14**-HSD120 (A) **LYS14**-GLU176 (A) **LYS14**-GLU180 (A) **ASP16**-ARG99 (A) **ASP16**-SER123 (A) **ARG17**-GLU176 (A) **SER22**-GLY117 (B) **LYS27**-GLU172 (B) **LYS27**-GLU176 (B) **LEU29**-TYR168 (B) **HSD32**-ARG165 (B)

Residues in L2 and BNP (annotated in bold) involved in the H-bond interactions allowing the best receptor–ligand stability were specifically determined using VMD software and visualized by PyMol at 3 Å. BNP, B-type natriuretic peptide; L2, Lebetin 2; NPR-A, natriuretic peptide receptor A; NPR-B, natriuretic peptide receptor B; NPR-C, natriuretic peptide receptor C. (A), chain A; (B) chain B; (C) chain C of natriuretic receptors.

**Table 5 toxins-14-00787-t005:** Number of electrostatic and hydrophobic interactions between natriuretic receptors and their ligands (L2 and BNP).

		hNPR-A		hNPR-B		hNPR-C	
	Type of Interaction	Monomer 1	Monomer 2	Monomer 1	Monomer 2	Monomer 1	Monomer 2
L2	Electrostatic	5	5	8	8	7	6
	Hydrophobic aromatic	1	2	1	1	1	2
	Hydrophobic non aromatic	4	5	5	1	3	4
BNP	Electrostatic	9	9	18	8	10	3
	Hydrophobic aromatic	-	-	-	-	1	-
	Hydrophobic non aromatic	12	5	3	6	1	4

Electrostatic and hydrophobic interactions between the two natriuretic receptor monomers and L2 and BNP were determined using Protein Structure and Interaction Analyzer (PSAIA) at 4.5 Å. BNP, B-type natriuretic peptide; L2, Lebetin 2; NPR, Natriuretic peptide receptor; hNPR-A, human natriuretic peptide receptor A; hNPR-B, human natriuretic peptide receptor B; hNPR-C, human natriuretic peptide receptor C.

**Table 6 toxins-14-00787-t006:** Representative table of electrostatic and hydrophobic interactions between natriuretic receptors and their ligands (L2 and BNP).

		hNPR-A	hNPR-B	hNPR-C			hNPR-A	hNPR-B	hNPR-C
L2	Monomer 1 electrostatic interactions	ASP2-ARG95(A) LYS18-ASP177(A) ASP20-ARG95(A) ARG21-TYR120(A) ARG21-GLU169(A)	ASP2-LYS110(A) LYS18-TYR159 (A) LYS18-GLU163(A) LYS18-GLU167(A) ASP20-LYS110(A) ARG21-TYR81(A) ARG21-SER85(A) SER26-TYR56(A)	LYS18-GLU176(A) LYS18-GLU180(A) ARG21-TYR125(A) ARG21-TYR168(A) ARG21-GLU172(A) SER26-LYS162(A) SER26-ARG165(A)	BNP	Monomer 1 electrostatic interactions	SER8-ARG178(A) ARG13-ASP177(A) LYS14-ASP177(A) ARG17-TYR120(A) ARG17-GLU169(A) LYS27-TYR154(A) LYS27-GLU187(A) ARG30-TYR154(A) ARG30-GLU169(A)	ARG13-SER172(A) LYS14-GLU129(A) LYS14-GLU167(A) LYS14-ASP361(A) LYS14-ASP365(A) SER21-SER85(A) SER21-ARG156(A) SER22-ARG156(A) LYS27-TYR148(A) LYS27-ASP150(A) LYS27-TYR159(A) LYS27-TYR181(A) ARG30-TYR148(A) ARG30-GLU163(A) ARG30-GLU167(A) ARG30-TYR148(A) ARG31-TYR148(A) ARG31-TYR159(A)	ARG13-ASP122(A) ARG13-GLU180(A) ASP16-ARG99(A) SER21-ARG165(A) SER22-ARG165(A) LYS27-TYR168(A) ARG30-TYR168(A) ARG30-GLU172(A) ARG30-GLU176(A) ARG31-TYR168 (A)
	Monomer 1 hydrophobic interactions	PHE15-PHE165(A) PRO11-MET173(A) CYS14-MET173(A) ILE19-VAL116(A) LEU28VAL59(A)	PHE15-PHE166(A) PRO5-LEU174(A) PRO6-LEU174(A) PRO11-VAL176(A) LEU28-PRO82(A) VAL33-CYS53(A)	PHE15-PHE169(A) LEU28-CYS62(A) LEU28-CYS90(A) LEU28-ALA93(A) LEU28-LEU163(A)		Monomer 1 hydrophobic interactions	PRO2-ALA138(A) PRO2-ALA139(A) PRO2-LEU179(A) CYS10-MET173(A) ILE18-VAL87(A) ILE18-ALA90(A) ILE18-ALA91(A) ILE18-ALA111(A) ILE24-ALA155(A) ILE24-PRO158(A) ILE24-ALA189(A) CYS26-ALA189(A)	PRO2-LEU95(A) MET15-ALA168(A) LEU29-VAL180(A)	PHE11-TRP347(A) CYS10-ALA362(A)
	Monomer 2 electrostatic interactions	LYS4-TYR88(B) LYS4-TYR156(B) LYS4-ASP160(B) LYS4-GLU161(B) LYS4-GLU162(B)	LYS4-TYR181(C) LYS7-GLU184(C) LYS8-TYR181(C) LYS8-GLU184(C) ASP20-ARG152(C) ARG21-GLU155(C) ARG21-TYR81(C) ARG21-ASP155(C)	LYS7-ASP195(B) LYS7-TYR188(B) LYS8-ASP191(B) LYS8-ASP197(B) LYS8-ASP200(B) ARG21-168TYR(B)		Monomer 2 electrostatic interactions	ARG17-ASP62(B) ARG17-TUR88(B) LYS27-TYR154(B) LYS27-GLU169(B) LYS27-GLU187(B) ARG31-TYR88(B) ARG31-TYR156(B) ARG31-GLU162(B) ARG31-GLU169(B)	ASP16-ARG152(C) ARG17-TYR81(C) ARG17-ASP150(C) ARG17-THR153(C) ARG17-ASP155(C) ARG31-TYR81(C) ARG31-ASP155(C) ARG31-TYR159(C)	ARG17-TYR92(B) ARG17-GLU164(B) SER21-ARG99(B)
	Monomer 2 hydrophobic interactions	PHE15-TYR154(B) PHE15-PHE165(B) PRO5-PRO158(B) PRO6-PRO158(B) PRO6-ALA189(B) CYS30-VAL116(B) CYS30-MET173(B)	PHE15-TYR181(C) ILE22-LEU149(C)	PHE15-TYR181(B) PHE15-PHE190(B) PRO10-187ILE(B) PRO10-207ALA(B) ILE22-163LEU(B)		Monomer 2 hydrophobic interactions	MET15-VAL59(B) MET15-PRO158(B) LEU24–MET173(B) VAL28-LEU186(B) LEU29-ALA155(B)	ILE18-CYS53(C) ILE18-CYS79(C) ILE18-VAL80(C) ILE18-PRO82(C) ILE18-ALA151(C) LEU29-LEU149(C)	VAL28-ILE187(B) LEU29-LEU156(B) LEU29-LEU171(B) LEU29-ILE187(B)

The residues involved in electrostatic and hydrophobic interaction between the two natriuretic receptor monomers (1 and 2) and L2/BNP were determined using Protein Structure and Interaction Analyzer (PSAIA) at 4.5 Å. BNP, B-type natriuretic peptide; L2, Lebetin 2; NPR-A, natriuretic peptide receptor A; NPR-B, natriuretic peptide receptor B; NPR-C, natriuretic peptide receptor C.

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
