# Peer review of "In Silico Study of the Mechanisms Underlying the Action of the Snake Natriuretic-Like Peptide Lebetin 2 during Cardiac Ischemia"

_toxins, 2022, doi:10.3390/toxins14110787_

Round 1

Reviewer 1 Report

In this study, the authors claimed that Lebetin 2 exhibited higher affinity for all human NPRs compared with BNP in all tested species by using computer simulation experiments. Although this is an interesting topic for the readers of the journal, a significant revision is needed. Below are more detailed questions that need to be addressed. 

1. Although the authors explain the selectivity of L2 to NRPs, only computer experiments are not enough and the authors needs to perform molecular interaction experiments (SPR, ITC, or MST) or point mutation experiments to prove the hypothesis. 

2. Protein modeling methods or software had a great influence on the simulation results. We suggest the authors test the other protein modeling further, like AlphaFlod. 

3. More detailed docking and molecular dynamics simulation parameters and protocols should be given.

Author Response

Response to Reviewer 1 Comments

Point 1: In this study, the authors claimed that Lebetin 2 exhibited higher affinity for all human NPRs compared with BNP in all tested species by using computer simulation experiments. Although this is an interesting topic for the readers of the journal, a significant revision is needed. Below are more detailed questions that need to be addressed. 

Although the authors explain the selectivity of L2 to NRPs, only computer experiments are not  enough and the authors needs to perform molecular interaction experiments (SPR, ITC, or MST) or point mutation experiments to prove the hypothesis. 

Response 1: Firstly, we thank the reviewer for the time spent reading and editing this document, and also for this very useful suggestion to perform molecular interaction experiments to prove our results. In the current work, we investigated the experimental hypothesis that L2 might exert its stronger and additional cardioprotective actions over BNP through better interaction with NPRs. Our in silico findings support this hypothesis and pave the way for further experimental exploration of the receptor/ligand interaction. In this context, we plan to use surface plasmon resonance (SPR) by systems such as BIACORE® to measure in real time, and without specific labelling, the characteristics of interaction between L2/BNP and their natriuretic receptors. We also plan to use NMR to further characterize this interaction. A single 2D-HSQC spectrum will be recorded from the NPRs and another one will be recorded from the NPRs supplemented with an excess of NPs in the same conditions, in the aim to determine the residues involved in the interaction. These experiments are planned as a working collaboration where all experimental aspects of the interaction will be extensively studied. However, these experiments cannot be presented in the current work because it takes time to perform such studies, and will therefore be published later.

Point 2: Protein modeling methods or software had a great influence on the simulation results. We suggest the authors test the other protein modeling further, like AlphaFlod.

Response 2: More and more online software are robust and efficient in determining 3D models like AlphaFold using artificial intelligence, but such software give us free ligand models while in this work we are interested on the structure of ligands in complex with their receptors. The explanation has been added in the text at the end of the “Results 2.1. Amino acid sequence and model analysis” section “Given the structural differences observed between the free forms of the ligands and the complex forms (Table 1) where the RMSD varies between 13 and 19 Å between the complex form of L2 and the free form, we decided to make 3D models of all the ligands used in the work based on the 3D structure of BNP in complex with NPR-C (PDB code 1YK1) and not to use the 3D models of the ligands available in the PDB or determined by alphaFold (Jumper, J., Nature, 2021)”.

We also started this work using the NMR structure of free L2 as a ligand for docking but the results gave us a low energy of interaction between all free used ligands and receptors. For this reason we decided to use MODELLER models in our work.

We also added the Table 1 below in the manuscript :

Table 1. RMSD of free L2 MODELLER model compared to 20 NMR models (PDB code 1Q01) and the free L2 AlphaFold model (Q7LZ09).

RMSD (Å)

L2 MODELLER model

AlphaFold_L2 model

16.576

1Q01_01 model

14.110

1Q01_02 model

18.277

1Q01_03 model

14.878

1Q01_04 model

17.960

1Q01_05 model

14.591

1Q01_06 model

16.407

1Q01_07 model

14.373

1Q01_08 model

17.637

1Q01_09 model

17.412

1Q01_10 model

16.558

1Q01_11 model

13.952

1Q01_12 model

16.115

1Q01_13 model

13.961

1Q01_14 model

14.061

1Q01_15 model

18.362

1Q01_16 model

13.592

1Q01_17 model

16.661

1Q01_18 model

19.069

1Q01_19 model

16.237

1Q01_20 model

14.835

Point 3: More detailed docking and molecular dynamics simulation parameters and protocols should be given.

Response 3: We agree with the suggestion of reviewer and have added method details in “Methods 5.1.5. Receptor-ligand molecular docking” section: “During the docking procedure, L2, ANP, BNP, CNP and DNP were used as ligands, with almost all of their bonds defined as no rotatable. All receptors were kept rigid. Grid maps representing the target proteins were constructed with different dimensions depending on the active site of the target protein. Autodock Tools software was used to prepare ligand and receptor files (pdbqt), attribute charges and calculate grid box. Based on the Autodock Tools software and during the preparation of the receptors, the protonation was carried out by adding only polar hydrogens, and applying the Kollman charges for the proteins, this is available in the “Edit” menu of the Autodock Tools software. For the ligands the Gasteiger charges were applied and all these treatments were carried out on the basis of the physiological pH. During docking, the parameters corresponding to the genetic-algorithm were set by default with ga_pop_size 150 (number of individuals in the population) and ga_num_evals 25000000 (maximum number of energy evaluations).”

In “Methods 5.2. Molecular dynamics simulation” section, we added “using Langevin dynamics with a damping constant of 1 ps−1. Energy minimization was performed for 1,000 steps. After the minimization step, the last frames were used to start MD simulations for 130ps using CHARMM36/NAMD. Only the complex which presents the strongest potential will be simulated in MD simulation for 100ns (Brooks et al., 2009; Phillips et al., 2005). The root mean square deviation (RMSD) and root mean square fluctuation (RMSF) were calculated using the VMD tools (Humphrey et al., 1996; Wong and Goscinski, 2012). The atomic charges of drugs studied in the dynamics simulation were assigned by adding parameter files in the dynamic script such as ligand.prm generated from charmm-gui  (corresponding to ligand) and other parameter files such as par_all36_prot.prm, par_all36_na.prm, par_all36_lipid.prm, par_all36_cgenff.prm, par_all36_carb.prm, par_all27_prot_lipid.inp, par_all22_prot.prm. Complete trajectories and analysis of receptor-ligand interactions were visualized in VMD.”

Reviewer 2 Report

The observation are inserted in pdf file.

Author Response

Response to Reviewer 2 Comments

Response 1: We thank the reviewer for the time spent reading and editing this manuscript.

As recommended by the reviewer, we inserted the required references in “Conclusions” section. The added references are [12, 13].

Reviewer 3 Report

The work “Lebetin 2, a snake venom-derived natriuretic peptide, as a cardioprotective agent: an in silico study” is devoted to the peptide (38-residue-long), Lebetin-2 (according to Fig.1’ amino acid sequence of the peptide: GDNKPPKKGPPNGCFGHKIDRIGSHSGLGCNKVDDNKG). The peptide is present in the venom of Macrovipera lebetina (Levantine viper). The amino-acid sequence of the peptide is available in the Uniprot.org databank under the code of Q7LZ09 and the name of Lebetin-2-alpha:

>sp|Q7LZ09|LEB_MACLB Lebetin-2-alpha OS=Macrovipera lebetina OX=8709 PE=1 SV=1 GDNKPPKKGPPNGCFGHKIDRIGSHSGLGCNKVDDNKG. The authors report on the interaction of this peptide with several receptors and compare these results with the respective data obtained for another peptide, BNP (or hBNP, according to Fig. 1) taken as a reference. The obtained data will help to develop clinically relevant therapeutic molecules for the treatment of ischemic heart disease, according to the authors. For Lebetin-2-alpha peptide in the Uniprot-databank as many as two spatial models are available. One is NMR-model (PDB-code: 1Q01) and another one, obtained with Alpha-fold software (Jumper et al., Nature 2021, 596(7873):583-589). However, the authors obtained their own model, using MODELLER software (dated by 2008, reference No48 in the Reference list). From this viewpoint it seems that the most problematic point of the work is the choice of the structural model for the peptide. Because, it is clear that the docking results will depend substantially on this choice. Therefore, the work will become more clear to the readers if the authors explain their choice for the structures of the NP peptides. Moreover, there are indications, judging from NMR model, that the peptide is rather flexible in aqueous solution and likely adopts its definite structure only upon interaction with the appropriate receptor. I think that all sources of the spatial data for Lebetin-2 should be discussed in the work and compelling reason is provided for choosing only one model obtained with MODELLER program. Especially, taken into account that the structure is flexible. I also do not understand from the work how the spatial structure of hBNP (or BNP) was obtained. An additional Figure comparing the NP’ spatial models would provide more clarity (maybe, we need to consider a broader list of NPs, available in Uniprot database, because their spatial models are solved by Alpha-fold). Probably, consideration of performing long-term MD-simulation of the chosen model in aqueous solution will provide more evidence for choosing a single one for the study its interaction with NPRs. Also, the work suffers from poor figure’ legends. Figure 1: the legend is not complete. What are the numbers in the last column? Probably, they reflect the lengths of the peptides. Please, indicate that clearly in the legend. Figure 2. The legend is too short. It is not clear where the receptor and where is the peptide. The colour coding is used in the Figure? What are orange spheres? Concerning Figures 3-5, they are difficult to understand. Probably, the interaction data could be organised as Tables, like Table S8? Some sentences are difficult to understand, e.g. in the text to Table 2: “i.e. the more the interaction energy between receptor and ligand is low…”. Probably, you mean: the lower the interaction energy…

line 289: “during 100ns MD simulations”. It sounds better: during 100 ns – long MD simulations.

I would recommend reading of the manuscript by a native speaker to improve its soundness.

Author Response

Response to Reviewer 3 Comments

The work “Lebetin 2, a snake venom-derived natriuretic peptide, as a cardioprotective agent: an in silico study” is devoted to the peptide (38-residue-long), Lebetin-2 (according to Fig.1’ amino acid sequence of the peptide: GDNKPPKKGPPNGCFGHKIDRIGSHSGLGCNKVDDNKG). The peptide is present in the venom of Macrovipera lebetina (Levantine viper). The amino-acid sequence of the peptide is available in the Uniprot.org databank under the code of Q7LZ09 and the name of Lebetin-2-alpha:

>sp|Q7LZ09|LEB_MACLB Lebetin-2-alpha OS=Macrovipera lebetina OX=8709 PE=1 SV=1 GDNKPPKKGPPNGCFGHKIDRIGSHSGLGCNKVDDNKG. The authors report on the interaction of this peptide with several receptors and compare these results with the respective data obtained for another peptide, BNP (or hBNP, according to Fig. 1) taken as a reference. The obtained data will help to develop clinically relevant therapeutic molecules for the treatment of ischemic heart disease, according to the authors. For Lebetin-2-alpha peptide in the Uniprot-databank as many as two spatial models are available. One is NMR-model (PDB-code: 1Q01) and another one, obtained with Alpha-fold software (Jumper et al., Nature 2021, 596(7873):583-589).

Point 1: However, the authors obtained their own model, using MODELLER software (dated by 2008, reference No48 in the Reference list). From this viewpoint it seems that the most problematic point of the work is the choice of the structural model for the peptide. Because, it is clear that the docking results will depend substantially on this choice. Therefore, the work will become more clear to the readers if the authors explain their choice for the structures of the NP peptides. Moreover, there are indications, judging from NMR model, that the peptide is rather flexible in aqueous solution and likely adopts its definite structure only upon interaction with the appropriate receptor. I think that all sources of the spatial data for Lebetin-2 should be discussed in the work and compelling reason is provided for choosing only one model obtained with MODELLER program. Especially, taken into account that the structure is flexible. I also do not understand from the work how the spatial structure of hBNP (or BNP) was obtained. An additional Figure comparing the NP’ spatial models would provide more clarity (maybe, we need to consider a broader list of NPs, available in Uniprot database, because their spatial models are solved by Alpha-fold). Probably, consideration of performing long-term MD-simulation of the chosen model in aqueous solution will provide more evidence for choosing a single one for the study its interaction with NPRs. 

Response 1: We thank the reviewer for the time spent reading and editing this manuscript and for the relevance of his comments and recommendations. Indeed more and more online software are robust and efficient in determining 3D models like AlphaFold using artificial intelligence, but such software give us free ligand models while in this work we are interested on the structure of ligands in complex with their receptors. The explanation was added in the manuscript at the end of the “Results 2.1. Amino acid sequence and model analysis” section “Given the structural differences observed between the free forms of the ligands and the complex forms (Table 1) where the RMSD varies between 13 and 19 Å between the complex form of L2 and the free form, we decided to make 3D models of all the ligands used in the work based on the 3D structure of BNP in complex with NPR-C (PDB code 1YK1) and not to use the 3D models of the ligands available in the PDB or determined by alphaFold (Jumper, J., Nature, 2021)”.

We also started this work using the NMR structure of free L2 as a ligand for docking but the results gave us a low interaction energy between all free ligands and receptors used. For this reason, we decided to use MODELLER models in our work.

We have also added in the manuscript Table 1 below:

Table 1. RMSD of free L2 MODELLER model compared to 20 NMR models (PDB code 1Q01) and the free L2 AlphaFold model (Q7LZ09).

RMSD (Å)

L2 MODELLER model

AlphaFold_L2 model

16.576

1Q01_01 model

14.110

1Q01_02 model

18.277

1Q01_03 model

14.878

1Q01_04 model

17.960

1Q01_05 model

14.591

1Q01_06 model

16.407

1Q01_07 model

14.373

1Q01_08 model

17.637

1Q01_09 model

17.412

1Q01_10 model

16.558

1Q01_11 model

13.952

1Q01_12 model

16.115

1Q01_13 model

13.961

1Q01_14 model

14.061

1Q01_15 model

18.362

1Q01_16 model

13.592

1Q01_17 model

16.661

1Q01_18 model

19.069

1Q01_19 model

16.237

1Q01_20 model

14.835

L2, Lebetin 2; NMR, nuclear magnetic resonance; RMSD; root mean square deviation.

All the models of the free ligands (NMR or AlphaFold or I-tasser structure) are drastically different from the models resulting from MODELLER with the structures of ligands in complexes. Moreover, dynamic simulations are not carried out on complexes derived from free ligands because the docking does not give interesting results.

To clarify this point, we have added to the attention of the reviewer an informative figure (see below). This figure shows the alignment of L2 with ANP, BNP, CNP and DNP whose models are obtained by MODELLER. The structural models of the free ligands given by AlphaFold are even more mobile and do not fit with the structure of the free form of L2. For this reason, this work was based only on the models obtained by MODELLER and reproducing the real interaction.

 Point 2: Also, the work suffers from poor figure’ legends.

Figure 1: the legend is not complete. What are the numbers in the last column? Probably, they reflect the lengths of the peptides. Please, indicate that clearly in the legend. 

Response 2.1. : The legend of Figure 1 was modified and added in the manuscript in the legend of the corresponding graph: “Multiple alignments of the amino acid sequences of lebetin 2 (L2) compared to amino acids sequence of human natriuretic peptides (NPs): Atrial natriuretic peptide (ANP), B-type natriuretic peptide (BNP) and C-type natriuretic peptide (CNP), and snake venom-derived dendroaspis natriuretic peptide (DNP). The referenced functional natriuretic amino acids residues are dashed in green. The conserved amino acids in all natriuretic peptides are dashed in green and yellow. All natriuretic peptides have a carboxylic C-termini and amidated N-termini. The numbers in the last column correspond to the length of the peptides.”

Figure 2. The legend is too short. It is not clear where the receptor and where is the peptide. The colour coding is used in the Figure? What are orange spheres?

Response 2.2.: The caption of Figure 2 has been modified and inserted into the manuscript in the legend for the corresponding graph “(a) The amino acids of L2, Phe15, Lys18, Arg21, Gly23, and Asp34 form hydrogen bonds with NPR-A His185 (B), Asp177 (A), Asp62 (B), Glu162 (B) and Arg175 (B) respectively. (b) The amino acids of BNP, Arg13, Lys14, Arg17, Ser21, Ser22, and Gly25 form hydrogen bonds with NPR-A Tyr156 (A), Glu162 (A), Glu 169 (A), Asp77(A), and Glu162 (B) respectively. (c) The amino acids of L2, Lys18, Asp20, Gly23, Ser24, LYS32, and Lys37 form hydrogen bonds with NPR-B Glu163 (A), Arg183 (C), Asp155 (C), Glu55 (C), Glu55 (A), and Asp338 (C) respectively. (d) The amino acids of BNP, Arg13, Met15, Arg17, and Ser19 form hydrogen bonds with NPR-B Glu129 (A), Glu167 (A), Asp155 (C) and Glu55 (C) respectively. (e) The amino acids of L2, Lys18, Asp20, Arg21, and Lys37 form hydrogen bonds with NPR-C Glu176 (A), Lys194 (B), Glu176 (A), and Asp105 (B) respectively. (f) The amino acids of BNP, Arg13, Lys14, Asp16, Arg17, and Ser22 form hydrogen bonds with NPR-C Glu180 (A), Glu1176 (A), Arg99 (A), Glu176 (A), and Gly117 (B) respectively. Lebetin 2 (L2) and human B-type natriuretic peptide (BNP) are represented in green. The potassium ion (K+) are in orange, NPR chain (A) in cyan blue and chain (B) in gray. L2 and BNP residues in purple are involved in interaction with NPR residues (in blue).

Concerning Figures 3-5, they are difficult to understand. Probably, the interaction data could be organised as Tables, like Table S8?

Response 2.3.: We agree with the reviewer : Figures 3, 4 and 5 were removed from the manuscript and replaced by Table 6 below:

Table 6. Representative table of electrostatic and hydrophobic interactions between natriuretic receptors and their ligands (L2, BNP)

hNPR-A

hNPR-B

hNPR-C

hNPR-A

hNPR-B

hNPR-C

L2

Monomer A

Electrostatic interactions

ASP2-ARG95(A)

LYS18-ASP177(A)

ASP20-ARG95(A)

ARG21-TYR120(A)

ARG21-GLU169(A)

ASP2-LYS110(A)

LYS18-TYR159 (A)

LYS18-GLU163(A)

LYS18-GLU167(A)

ASP20-LYS110(A)

ARG21-TYR81(A)

ARG21-SER85(A)

SER26-TYR56(A)

LYS18-GLU176(A)

LYS18-GLU180(A)

ARG21-TYR125(A)

ARG21-TYR168(A)

ARG21-GLU172(A)

SER26-LYS162(A)

SER26-ARG165(A)

BNP

Monomer A

electrostatic interactions

SER8-ARG178(A)

ARG13-ASP177(A)

LYS14-ASP177(A)

ARG17-TYR120(A)

ARG17-GLU169(A)

LYS27-TYR154(A)

LYS27-GLU187(A)

ARG30-TYR154(A)

ARG30-GLU169(A)

ARG13-SER172(A)

LYS14-GLU129(A)

LYS14-GLU167(A)

LYS14-ASP361(A)

LYS14-ASP365(A)

SER21-SER85(A)

SER21-ARG156(A)

SER22-ARG156(A)

LYS27-TYR148(A)

LYS27-ASP150(A)

LYS27-TYR159(A)

LYS27-TYR181(A)

ARG30-TYR148(A)

ARG30-GLU163(A)

ARG30-GLU167(A)

ARG30-TYR148(A)

ARG31-TYR148(A)

ARG31-TYR159(A)

ARG13-ASP122(A)

ARG13-GLU180(A)

ASP16-ARG99(A)

SER21-ARG165(A)

SER22-ARG165(A)

LYS27-TYR168(A)

ARG30-TYR168(A)

ARG30-GLU172(A)

ARG30-GLU176(A)

ARG31-TYR168 (A)

Monomer A

hydrophobic interactions

PHE15-PHE165(A)

PRO11-MET173(A)

CYS14-MET173(A)

ILE19-VAL116(A)

LEU28VAL59(A)

PHE15-PHE166(A)

PRO5-LEU174(A)

PRO6-LEU174(A)

PRO11-VAL176(A)

LEU28-PRO82(A)

VAL33-CYS53(A)

PHE15-PHE169(A)

LEU28-CYS62(A)

LEU28-CYS90(A)

LEU28-ALA93(A)

LEU28-LEU163(A)

Monomer A

hydrophobic interactions

PRO2-ALA138(A)

PRO2-ALA139(A)

PRO2-LEU179(A)

CYS10-MET173(A)

ILE18-VAL87(A)

ILE18-ALA90(A)

ILE18-ALA91(A)

ILE18-ALA111(A)

ILE24-ALA155(A)

ILE24-PRO158(A)

ILE24-ALA189(A)

CYS26-ALA189(A)

PRO2-LEU95(A)

MET15-ALA168(A)

LEU29-VAL180(A)

PHE11-TRP347(A)

CYS10-ALA362(A)

Monomer B

electrostatic interactions

LYS4-TYR88(B)

LYS4-TYR156(B)

LYS4-ASP160(B)

LYS4-GLU161(B)

LYS4-GLU162(B)

LYS4-TYR181(C)

LYS7-GLU184(C)

LYS8-TYR181(C)

LYS8-GLU184(C)

ASP20-ARG152(C)

ARG21-GLU155(C)

ARG21-TYR81(C)

ARG21-ASP155(C)

LYS7-ASP195(B)

LYS7-TYR188(B)

LYS8-ASP191(B)

LYS8-ASP197(B)

LYS8-ASP200(B)

ARG21-168TYR(B)

Monomer B

electrostatic interactions

ARG17-ASP62(B)

ARG17-TUR88(B)

LYS27-TYR154(B)

LYS27-GLU169(B)

LYS27-GLU187(B)

ARG31-TYR88(B)

ARG31-TYR156(B)

ARG31-GLU162(B)

ARG31-GLU169(B)

ASP16-ARG152(C)

ARG17-TYR81(C)

ARG17-ASP150(C)

ARG17-THR153(C)

ARG17-ASP155(C)

ARG31-TYR81(C)

ARG31-ASP155(C)

ARG31-TYR159(C)

ARG17-TYR92(B)

ARG17-GLU164(B)

SER21-ARG99(B)

Monomer B

hydrophobic interactions

PHE15-TYR154(B)

PHE15-PHE165(B)

PRO5-PRO158(B)

PRO6-PRO158(B)

PRO6-ALA189(B)

CYS30-VAL116(B)

CYS30-MET173(B)

PHE15-TYR181(C)

ILE22-LEU149(C)

PHE15-TYR181(B)

PHE15-PHE190(B)

PRO10-187ILE(B)

PRO10-207ALA(B)

ILE22-163LEU(B)

Monomer B

hydrophobic interactions

MET15-VAL59(B)

MET15-PRO158(B)

LEU24–MET173(B)

VAL28-LEU186(B)

LEU29-ALA155(B)

ILE18-CYS53(C)

ILE18-CYS79(C)

ILE18-VAL80(C)

ILE18-PRO82(C)

ILE18-ALA151(C)

LEU29-LEU149(C)

VAL28-ILE187(B)

LEU29-LEU156(B)

LEU29-LEU171(B)

LEU29-ILE187(B)

The residues involved in electrostatic and hydrophobic interaction between the two natriuretic receptor monomers and L2/BNP were determined using Protein Structure and Interaction Analyzer (PSAIA) at 4.5 Å. BNP, B-type natriuretic peptide ; L2, Lebetin 2 ; NPR-A, natriuretic peptide receptor A ; NPR-B, natriuretic peptide receptor B ; NPR-C, natriuretic peptide receptor C.

Point 3: Some sentences are difficult to understand, e.g. in the text to Table 2: “i.e. the more the interaction energy between receptor and ligand is low…”. Probably, you mean: the lower the interaction energy…

line 289: “during 100ns MD simulations”. It sounds better: during 100 ns – long MD simulations.

I would recommend reading of the manuscript by a native speaker to improve its soundness.

Response 3: As suggested by the reviewer, grammatical errors have been corrected throughout the text.

The manuscript was also proofread and corrected for the improvement of the English.

Round 2

Reviewer 1 Report

The authors have made great effort to improve the quality of the manuscript and answered most of the questions.  I have no more questions.

Author Response

Response to Reviewer 1 Comments

We thank the reviewer for his time and effort in reviewing our manuscript. The feedback has been invaluable in improving the content and presentation of the paper. We sincerely appreciate all your valuable comments and suggestions, which helped us in improving the quality of the manuscript. We hope that the revised paper is now suitable for inclusion in Toxins.

Sincerely,

Reviewer 3 Report

If there is no difference between L2 and Lebetin-2-alpha (Q7LZ09), please add the phrase To Fig. 1 legend (or Introduction) that L2 is identical L2-alpha. 

Author Response

Response to Reviewer 3 Comments

We thank the reviewer for his time and effort in reviewing our manuscript. The feedback has been invaluable in improving the content and presentation of the paper. We sincerely appreciate all your valuable comments and suggestions, which helped us in improving the quality of the manuscript.

In accordance with the reviewer´s comment, we have responded to point 1 below, and the entire manuscript has undergone extensive substantial English editing.

We hope that the revised paper is now suitable for inclusion in Toxins.

Sincerely,

Point 1 : If there is no difference between L2 and Lebetin-2-alpha (Q7LZ09), please add the phrase To Fig. 1 legend (or Introduction) that L2 is identical L2-alpha. 

Response 1: We thank the reviewer for this 2nd revision and the positive comment. As suggested, we modified the legend of Figure 1 and replaced L2 with Lebetin-2-alpha (Q7LZ09), and also clarified this point in the Introduction section as follows « As the cardioprotective effect of L2 has been reported for the L2-alpha isoform (38 amino acids) [12, 13], we used in this study L2-alpha-Q7LZ09 whose structure coordinate was determined by nuclear magnetic resonance (NMR) spectroscopy, and also available in the Protein Data base (PDB) (PDB ID: 1Q01) [24].»
